# Patterns of postnatal weight gain and its predictors among preterm very low birth weight neonates born in Bahir-Dar city public hospitals, 2022: A cross sectional study

**Yihenew Ayehu Dessie**[1]*, **Worku Abemie**[2], **Elda Mekonnen Nigussie**[3], **Bethelehem Taye Mengistu**[3], **Leweyehu Alemaw Mengstie**[2], **Bekahegn Girma**[2], **Solomon Hailemeskel**[4]

1 Department of Pediatrics and Child Health Nursing, School of Nursing and Midwifery, Institute of Health, Bule Hora University, Bule Hora, Ethiopia, 2 Department of Pediatrics and Child Health Nursing, School of Nursing and Midwifery, Asrat Woldeyes Health Sciences Campus, Debre Berhan University, Debre Berhan, Ethiopia, 3 Department of Nursing, School of Nursing and Midwifery, Asrat Woldeyes Health Sciences Campus, Debre Berhan University, Debre Berhan, Ethiopia, 4 Department of Midwifery, School of Nursing and Midwifery, College of Health Sciences, Debre Berhan University, Debre Berhan, Ethiopia

* filimonayehu21@gmail.com

**Data Availability Statement:** The datasets analyzed in this study are available on supplementary file.

## Abstract

### Introduction

Postnatal weight gain in very low birth weight infants remains a challenge during the neonatal period in low and middle-income countries like Ethiopia, where no feeding alternatives and follow-up charts are available. Although extrauterine growth retardation is a common problem in preterm very low birth weight infants, there is a lack of evidence in resource-limited countries regarding patterns of postnatal weight gain. Therefore, this study aimed to assess the patterns of postnatal weight gain and its predictors among preterm very low birth weight infants in Ethiopia.

### Methods

A cross-sectional study was conducted on a randomly selected sample of 412 neonates in Ethiopia. Data were collected using structured questionnaires and analyzed with Stata version 14.0 software. Bivariable and multivariable logistic regression analyses were performed to identify significant predictors. Model fitness and assumptions were assessed. Associations were reported using adjusted odds ratios (AOR) with 95% confidence intervals.

### Results

In the current study, 14.6% (95% CI: 10.4–20.1) of neonates had adequate postnatal weight gain at discharge. Spontaneous vaginal delivery [AOR: 2.54; 95% CI (1.17, 5.54)], birth Z-score > -1.29 [AOR: 4.51; 95% CI (1.43, 14.16)], early feeding initiation time [AOR: 3.36; 95% CI (1.63, 6.92)], and respiratory distress syndrome [AOR: 0.31; 95% CI (0.12, 0.78)]

**Funding:** The authors also confirmed that no financial funding was received for the study, authorship, or publication.

**Competing interests:** The authors declare that there is no competing interest.

**Abbreviations:** AOR, adjusted odd's ratio; BW, birth weight; CI, confidence interval; NEC, necrotizing enterocolitis; NICU, neonatal intensive care unit; RDS, Respiratory Disease Syndrome; SVD, spontaneous vaginal delivery; VLBW, very low birth weight.

were significant predictors for postnatal weight gain among very low birth weight neonates in Ethiopia.

## Conclusion

The postnatal weight gain reported in this study was low as compared to the national figure. Mode of delivery, birth z-score, initiation time of the first feeding, and respiratory distress syndrome were associated with postnatal weight gain. The Federal Ministry of Health, stakeholders, national neonatal associations, and non-governmental organizations should work collaboratively to promote vaginal delivery and early initiation of feeding and develop guidelines specifically tailored for this special population. Furthermore, healthcare providers should prioritize and focus on neonates who have respiratory distress and low Z-scores.

## Background

The World Health Organization (WHO) defines a very low birth weight (VLBW) preterm infant as one born weighing less than 1500 grams and before 37 completed weeks of gestation [1, 2]. Postnatal weight gain in preterm VLBW newborns typically involves an initial weight loss of 7–15% during the first week, followed by recovery and weight increase from the 10th to the 21st day, measured in grams per kilogram per day after regaining birth weight [3, 4].

Neonates with birth weight less than 1500 g were at 49%, 70%, 80% increased odds of mortality compared to those 1500–2449 g, 2500–3999 g and more than 4000 g respectively [5]. In low-and middle-income countries (LMICs), premature VLBW infants who survive neonatal, and infancy periods have estimated prevalence of any neurodevelopmental impairment, cognitive impairment, and cerebral palsy of 21.4%, 16.3% and 11.2% respectively [6, 7]. In Ethiopia, the rates of neonatal mortality are still significantly high, as a prospective cohort study indicated that 29% of preterm neonates admitted to neonatal intensive care units (NICUs) do not survive, and among those who do survive, 86.2% faced growth restrictions at the time of their discharge from the hospital [8]. VLBW infants are at risk of poor postnatal weight gain due to inadequate nutrition in the first weeks of life [9].

Preterm VLBW infants are at increased risk of postnatal growth failure, particularly immediately after birth when they are most medically fragile and often transition slowly to enteral feeding [10]. Approximately 90% of VLBW infants are classified as growth restricted and falling below the 10th percentile on a standardized intrauterine growth curve due to poor postnatal weight gain of less than 15 g/kg/day by 36 weeks gestational age, which is a sign of delayed growth [11, 12]. Despite advancements in neonatal care, inadequate growth of VLBW infants remains a significant challenge [13]. These difficulties may rise from metabolic and gastrointestinal immaturity, a compromised immune system, and other medical complications such as necrotizing enterocolitis (NEC), respiratory distress syndrome (RDS), and sepsis [9, 13].

The postnatal weight gain of a preterm VLBW infant is influenced by the extent of intrauterine growth restriction, early postnatal weight loss, and the attention given by practitioners to ensure adequate weight gain [4]. Each preterm VLBW infant has unique maternal, fetal, nutritional, genetic, and environmental factors influencing growth [14]. Due to these varying factors, using a standard growth rate to define optimal growth and applying intrauterine growth rates from the third trimester as a standard for extrauterine growth is conceptually flawed [9].

The nutritional requirements for adequate weight gain in preterm VLBW infants are significantly influenced by the type, amount, and frequency of feeding during initial hospitalization, often based on expert opinion targeting a growth standard that may not be optimal [10]. Therefore, careful attention to early nutritional practices can minimize postnatal weight loss and ensure adequate postnatal weight gain [11].

Early detection of growth failure among preterm (VLBW) neonates is challenging in Ethiopia due to the lack of population-specific growth charts. Understanding these patterns will aid in developing and enhancing guidelines for managing VLBW infants in Ethiopia and similar settings. Despite the high burden of poor weight gain in resource-limited countries, evidence on postnatal growth patterns and predictors for VLBW preterm infants in Ethiopia remains limited. Therefore, this study aimed to address this gap by plotting growth curves for each gestational age and assessing postnatal weight gain patterns and predictors among preterm VLBW neonates in Bahir Dar public hospitals in Ethiopia.

## Methods

### Study setting and period

The research was conducted in public hospitals situated in Bahir Dar city, Amhara Region, Ethiopia. Bahir Dar city is home to three public hospitals: Tibebe Giwon, Felege Hiwot, and Adissalem. Each hospital's NICU has three sub-units designated for specific patient groups: one for term infants, one for critically preterm infants, and one for stable preterm infants in kangaroo mother care (KMC). The feeding practices in these hospitals were in accordance with the national guidelines of Ethiopia [15], which are ordered by physicians based on medical conditions and feeding tolerance. Nurses fed unstable preterm neonates via nasogastric (NG) tubes and instructed mothers on how to feed their stable preterm and term infants every three hours, following hospital guidelines. The study was conducted from April 2020 to April 2022.

**Study design.**   A facility-based cross sectional study was employed.

### Study population

All preterm VLBW neonates (born before 37 weeks of gestation with a birth weight of less than 1500 grams) admitted to the neonatal intensive care units of Bahir Dar city's public hospitals from April 2020 to April 2022.

### Eligibility criteria

Neonates born before 37 weeks of gestation with a birth weight of less than 1500 grams were included. However, those who were part of multiple births, had congenital malformations or chromosomal disorders diagnosed at birth, or had died or been discharged within 7 days before physiological weight gain began were excluded from the study.

**Sample size determination.**   The sample size was determined using the single population proportion formula, based on the following assumptions: a prevalence (p) of adequate postnatal weight gain among preterm VLBW (50%), since there were no studies done in Ethiopia, a confidence level (Zα/2) of 95%, and a margin of error of 5%.

$$n = \frac{\left(\frac{Z_{\frac{\alpha}{2}}}{}\right)^2 P(1-P)}{d^2}$$

After adding 5% non-response rate, the minimum required sample size was 423.

**Sampling technique and procedure.**   A systematic random sampling technique was employed to select the medical records of preterm VLBW neonates from each hospital. The

total number of preterm VLBW neonates admitted to the NICU during the data collection period was estimated from the registration book. Over the two-year period, the average number of preterm VLBW neonates was 360 in Tibebe Giwon, 301 in Felege Hiwot, and 192 in Adissalem. The number of records to be included in the study from each hospital was determined using the proportionate to size (PS) allocation technique. The sampling interval (k) was determined by dividing the expected number of patients by the calculated sample size, resulting in k = 2. The first chart, chosen by lottery and being the second chart, led to selecting every second eligible neonate's record according to the order of admission.

## Data collection

A structured questionnaire adapted from different studies was used to collect data from the preterm neonate cards [8, 13, 16, 17]. A pretest was conducted on 5% of the sample (22 charts) to ensure consistency across the recorded data, with necessary adjustments made based on the findings. Due to incomplete information on antenatal risk factors like medication history, maternal education, feeding habits, and health status, these factors were excluded from the final data collection tool. The data collection tools had four main parts: socio-demographic characteristics, feeding practices of the neonate, obstetric characteristics, and comorbid conditions of the neonate. The daily weight of each neonate until discharge was recorded. Weight gain from the time of regaining birth weight to discharge (growth velocity) was calculated by using the two-point average model. $GV = \frac{[1000*(wn-w1)]}{\left[(Dn-D1)*\frac{(wn+w1)}{2}\right]}$, where w = weight in grams, D = day, 1 = beginning of time interval, and n = end of time interval in days [18, 19].

Z-scores were calculated using the difference between the birth weight of the neonate and the weight of the reference neonate of the same sex and age obtained from intergrowth 21st divided by SD.

$$Z - score\ at\ birth = \frac{(birth\ weight\ of\ the\ neonate - weight\ of\ the\ neonate\ of\ the\ standard\ population)}{SD}$$

Similarly,

$$Z\text{-}score\ at\ discharge = \frac{(dischare\ weight\ of\ the\ neonate - weight\ of\ the\ neonate\ of\ the\ standard\ population)}{SD}$$

Data collection was carried out by four trained BSc nurses under the supervision of two MSc neonatal nurses from June 12 to July 20, 2022. To prevent data collector bias and ensure data quality, all data collectors and supervisors received training prior to the data collection period. Throughout the data collection process, regular supervision was maintained, and each chart was checked for consistency, completeness, and appropriate documentation.

## Operational definitions

**Small for gestational age:** birth weight-for-gestational-age measure below the 10th percentile compared to a sex-specific reference population [11].

**Postnatal growth failure** (PGF) was defined as a decrease in weight Z score between birth and discharge of more than −1.29 or less than 10 percentiles, using the Inter growth 21st standards and Fenton growth charts [20].

**Adequate postnatal weight gain:** average weight gain (g/kg/day) ≥15 g/kg/day [11, 16, 21].

**Poor postnatal weight gain:** average weight gain (g/kg/day) < 15 g/kg/day [11, 16, 21].

**Average weight loss;** the percentage of weight lost during the first week of life, calculated as [(birth weight—lowest weight in the first week) / birth weight] x 100.

**Early preterm neonates:** neonates born alive before 34 completed weeks of gestation [22].

**Late preterm neonates:** neonates born alive between 34 and 36 weeks of gestation [22, 23].

## Data management and analysis

Data were collected, cleaned, coded, entered using the ODK mobile application, and exported to Stata version 14.1 for analysis. Descriptive statistics were used to summarize the data and characterize the infants in the sample. Mean and standard deviation (SD) were used to describe continuous variables, and percentages were used for categorical variables. The kernel density curve and histogram graph show a normal distribution with residuals. The 3rd,10th,50th, 90th and 97th percentiles were calculated each day among samples, smoothed with a spline smoothing package in R, and plotted in a cubic spline smoothing graph within each gestational age from 29 to 36 weeks.

The outcome variable, average weight gain (g/kg/ day), was dichotomized as weight gain ≥15 g/kg/day (adequate weight gain) and weight gain < 15 g/kg/day (poor weight gain).

A bivariable and multivariate logistic regression model was used to assess factors associated with postnatal weight gain. An adjusted odds ratio (AOR) with 95% confidence intervals (CI) was used to show the association. To check for multicollinearity, the variance inflation factor (VIF) test after regression was used. Lastly, the model fitness was assessed by using Hosmer-Lemeshow's test.

## Ethics approval and consent to participate

This study adhered to the ethical principles of the Declaration of Helsinki. Ethical approval (IRB-013) was obtained from the Institutional Review Board at Debre Berhan University's Asrat Woldeyes Health Science campus. Formal letters were sent to hospital officials to secure their cooperation and permissions. Data were collected from neonate charts with patient cards assigned unique codes to ensure confidentiality. Only designated collectors accessed these codes, minimizing participant identification risks. Data were anonymized during analysis and reported in aggregate to protect participant privacy.

## Results

In the current study, a total of 412 preterm VLBW neonates were eligible and included with a response rate of 97.4%.

## Socio-demographic and obstetric characteristics

Among neonates included in the present study, two hundred twenty four (54.4%) of them were male. One hundred forty four (35%) of neonates were delivered by caesarean section, and 76.7% were in KMC units. Three hundred eighty four (93.2%) neonates were delivered by mothers who had antenatal care (ANC) follow-up. Two hundred thirty eight (57.8%) of the neonates were classified as early preterm VLBW neonates (Table 1).

## Time to regain birth weight; reach full volume feeds and average postnatal weight gain of preterm VLBW neonates

The average postnatal weight gain among preterm VLBW infants from birth to discharge was 5.5 ± 3.2 g/kg/day, while it was 11.1 ± 3.8 g/kg/day from the time of regaining birth weight to discharge. On average, these infants took 13.3 ± 5.3 days to regain their birth weight. The mean birth weight of the neonates was 1333.5 ± 140 g, and by the end of the first week, their mean weight decreased to 1219.6 ± 154.4 g, indicating an average weight loss of 8.7% during this period. By discharge, the average weight increased to 1513 ± 156 g (Table 2). The number

**Table 1. Socio-demographics and obstetric characteristics of preterm VLBW neonates and their mothers in Bahir Dar city public hospitals, 2022 (N = 412).**

| Variable | Category | Postnatal weight gain | | Total number | Frequency (%) |
|---|---|---|---|---|---|
| | | Adequate | Poor | | |
| Sex | Male | 30 | 158 | 188 | 45.6 |
| | Female | 30 | 194 | 224 | 54.4 |
| Birth weight in g | < = 1000 | 6 | 12 | 18 | 4.4 |
| | 1000–1100 | 10 | 8 | 18 | 4.4 |
| | 1100–1200 | 10 | 32 | 42 | 10.2 |
| | 1200–1300 | 16 | 74 | 90 | 21.8 |
| | 1300–1400 | 14 | 90 | 104 | 25.2 |
| | 1400–1500 | 4 | 136 | 140 | 34 |
| Mode of delivery | c- section | 12 | 132 | 144 | 35 |
| | SVD | 48 | 220 | 268 | 65 |
| Gestational age | early preterm | 24 | 214 | 238 | 57.8 |
| | late preterm | 36 | 138 | 174 | 42.2 |
| Size for GA | SGA | 16 | 156 | 172 | 41.8 |
| | AGA | 44 | 196 | 240 | 58.2 |
| Maternal age | 18–30 | 34 | 210 | 244 | 59.2 |
| | >30 | 26 | 142 | 168 | 40.8 |
| Number of children that the mother has | < = 1 | 18 | 140 | 158 | 38.4 |
| | 2 | 6 | 48 | 54 | 13 |
| | > = 3 | 36 | 164 | 200 | 48.6 |
| ANC visit | yes | 52 | 332 | 192 | 93.2 |
| | no | 8 | 20 | 28 | 6.8 |
| No. of ANC visit | < = 2 times | 26 | 174 | 200 | 52.1 |
| | >2 times | 26 | 158 | 184 | 47.9 |

**Table 2. Time to regain birth weight, reach full volume feeds, and average postnatal weight gain of preterm VLBW neonates based on gestational age in Bahir Dar city public hospitals, 2022 (N = 412).**

| GA | N | SGA N (%) | BW(g) Mean ± SD | Time to regain BW (days) Mean ±SD | Time to reach full feed (days) Mean ± SD | Average weight gain from birth to discharge in g/kg/d | Average weight gain from gaining BW to discharge in g/kg/d |
|---|---|---|---|---|---|---|---|
| ≤29 weeks | 48 | 32(66.7) | 1212.9±135 | 18.7±4.7 | 16.8±5.6 | 3.9±3.3 | 9.4±5 |
| 30 weeks | 30 | 6(20) | 1233.7±128.9 | 12.3±4.3 | 13.1±4.3 | 5.1±4 | 9±4.3 |
| 31 weeks | 58 | 20(34.5) | 1335.3±127.1 | 12.9±4.3 | 12.8±3.8 | 5±2.7 | 10.1±3.7 |
| 32 weeks | 40 | 12(30) | 1316.5±91.9 | 14±2.9 | 14.6±5.4 | 5.4±2.9 | 12±2.8 |
| 33 weeks | 62 | 24(38.7) | 1326.9±144.9 | 13.8±3.6 | 16.2±3 | 5.1±3.4 | 10.7±3.6 |
| 34 weeks | 54 | 32(59.3) | 1380.6±88.6 | 13.6±5.1 | 14.2±5 | 6±2.8 | 13.6±2.7 |
| 35 weeks | 68 | 30(44.1) | 1360±162.4 | 10.4±3.6 | 10.6±3.4 | 7±2.7 | 13±3.4 |
| 36 weeks | 52 | 16(30.8) | 1438.3±77.3 | 11.8±7.2 | 11.3±4.7 | 5.4±2.9 | 11.4±3.5 |
| Gestational age category | | | | | | | |
| Early preterm | 238 | 94(39.5) | 1292±136 | 14.4±4.9 | 16.9±4.6 | 4.9±3.4 | 10.3±3.9 |
| Late preterm | 174 | 78(44.8) | 1389±123 | 11.8±5.5 | 12±4.6 | 6.2±2.8 | 12.3±3.3 |
| Mean weight gain | | | | | | 5.5±3.2 | 11.1±3.8 |

**Table 3. Postnatal weight loss and Z-score in preterm VLBW neonates in Bahir Dar city public hospitals, 2022 (N = 412).**

| Variable | Category | Postnatal weight gain | | N | % | p-value |
|---|---|---|---|---|---|---|
| | | Adequate | Poor | | | |
| Average wt loss (%) | < = 10 | 38 | 228 | 266 | 64.6 | 0.879 |
| | >10 | 22 | 124 | 146 | 35.4 | |
| Time to regain birth wt | < = 14 days | 48 | 188 | 236 | 57.3 | 0.006 |
| | >14 days | 12 | 164 | 176 | 42.7 | |
| At birth Z-score | < = -1.29 | 26 | 274 | 300 | 2.8 | 0.000 |
| | >-1.29 | 34 | 78 | 112 | 27.2 | |
| At discharge Z-score | < = -1.29 | 8 | 40 | 48 | 11.7 | 0.660 |
| | >-1.29 | 12 | 312 | 364 | 88.3 | |

of neonates included for growth curve plotting decreased with the number of postnatal days, especially after day 18 due to discharge. Additional details are available in S1–S8 Tables in S1 File.

## Postnatal weight loss and Z-score in preterm VLBW neonates

The mean z-score of body weight at birth and discharge was -0.78 and -0.93 respectively. At discharge, nearly 12% of the neonates were considered to have growth failure. Over one-third (35.4%) of the neonates experienced more than 10% weight loss and 42.7% regained their birth weight 2 weeks after birth (Table 3).

Overall, 352 out of 412 (85.4%) CI: (80–89.7) of neonates had poor postnatal weight gain. Among these, (51.9%) were early preterm VLBW neonates (Fig 1).

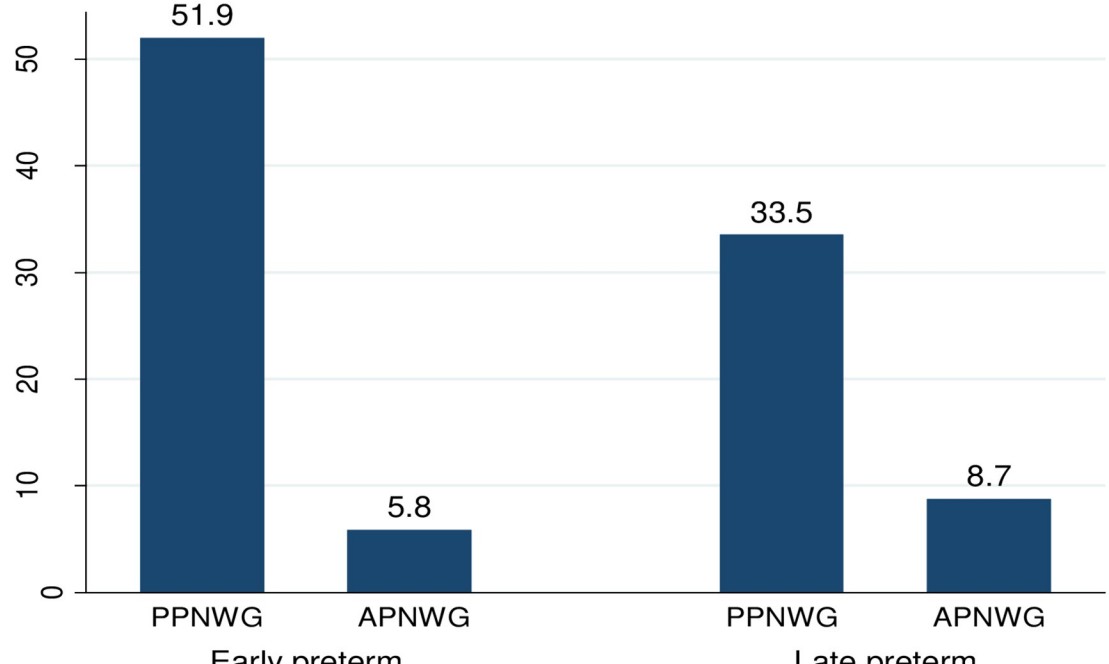

PPNWG=Poor postnatal weight gain and APNWG= Adequate postnatal weight gain

**Fig 1. Percentages of postnatal weight gain among early preterm and late preterm VLBW neonates.**

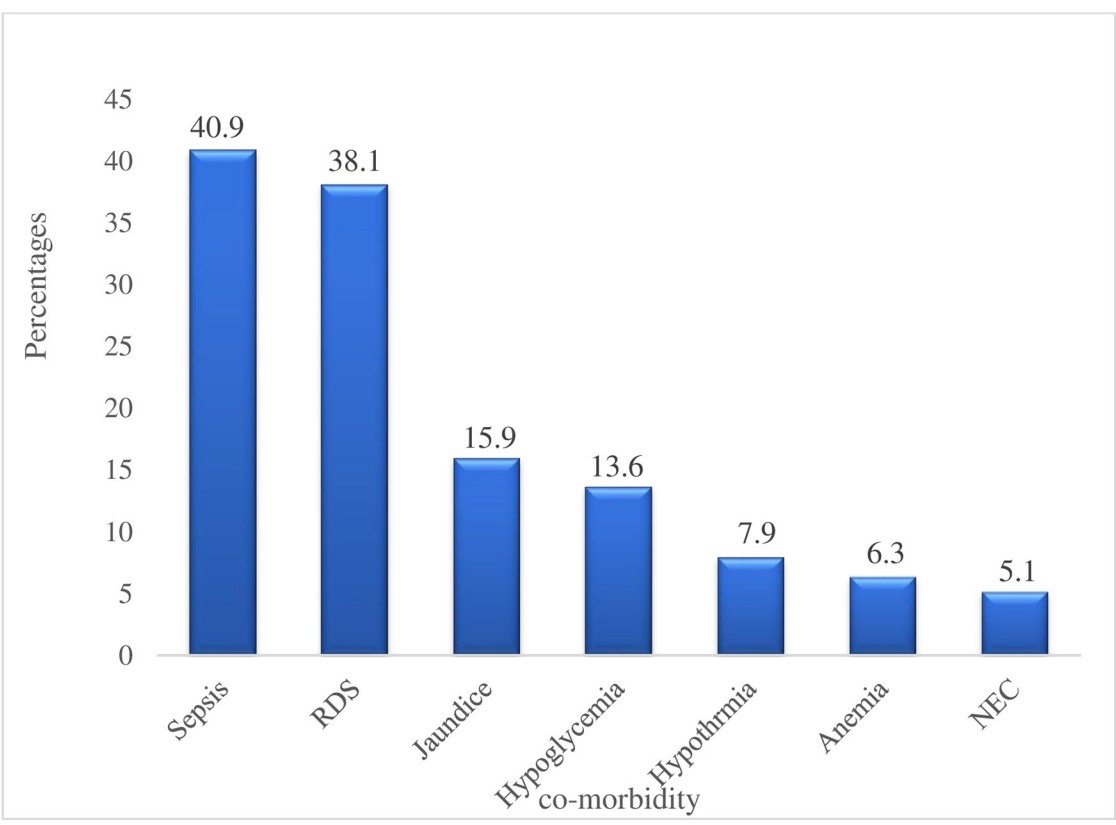

**Fig 2. Comorbidity conditions among preterm VLBW neonates with poor postnatal weight gain.**

**Variables related to hospitalization and comorbidities.** Among the total preterm VLBW neonates, the majority (60.2%) stayed for more than 22 days, 59.2% were on oxygen for more than 9 days, and 14.1% died. Of those who died, more than 93% of neonates had poor postnatal weight gain. Among preterm VLBW neonates who had poor postnatal weight gain, 132/352 (37.5%) had multiple comorbidities. Common comorbid conditions were sepsis, 144/352(40.9%), RDS 133/352(38.1%), and others as shown in (Fig 2).

**Feeding practice in preterm VLBW neonates.** Regarding feeding practice, most preterm VLBW neonates initiated their first feeding on the second day of life, receiving preferably breast milk via NG-tube every 3 hrs (Table 4).

**Predictors of postnatal weight gain among preterm VLBW neonates.** In bi-variable binary logistic regression analysis, the mode of delivery, gestational age, birth z-score, time of initiation of the first feeding, initial volume, frequency of feeding, time to regain birth weight, the maximum volume of feeding by the neonate, length of hospital stay, and respiratory distress syndrome were significantly associated with postnatal weight gain in preterm VLBW neonates. Further, these variables were analyzed in the multi-variable analysis mode of delivery, birth z-score, feeding initiation time and respiratory distress syndrome were significantly associated with postnatal weight gain.

Preterm VLBW neonates delivered by spontaneous vaginal delivery were 2.54.times (AOR = 2.54, 95% CI: (1.17–5.54)) more likely to have adequate postnatal weight gain than those who delivered by C-section, and Neonates with a birth z-score greater than -1.29 were 4.51 times (AOR = 4.51, 95% CI: (1.43–14.16)) more likely to have adequate postnatal weight gain compared to those with a birth z-score of ≤-1.29.

**Table 4. Feeding practice in preterm VLBW neonates in Bahir Dar city public hospitals, 2022 (N = 412).**

| Variable | Category | Postnatal weight gain | | N | % | p-value |
|---|---|---|---|---|---|---|
| | | Adequate | Poor | | | |
| Type of milk feed | breast milk | 59 | 205 | 364 | 88.4 | 0.287 |
| | formula milk | - | 14 | 14 | 3.4 | |
| | Mixed | 2 | 32 | 34 | 8.2 | |
| Frequency of feeding | ≤ every 3 hr | 56 | 298 | 354 | 85.9 | 0.017 |
| | > every 3 hr | 4 | 54 | 58 | 14.1 | |
| Initial method of feeding | tube feeding | 59 | 349 | 408 | 99 | 0.557 |
| | breast feeding | - | 4 | 4 | 1 | |
| Time of initiation | On 1st day | 44 | 126 | 170 | 41.3 | 0.000 |
| | On 2nd day | 16 | 226 | 242 | 58.7 | |
| Initial volume | <2ml | 7 | 53 | 60 | 14.6 | 0.012 |
| | 2ml | 35 | 163 | 298 | 72.7 | |
| | >2ml | 18 | 36 | 54 | 12.7 | |
| Maximum volume to reach full feeding | <150ml | 10 | 118 | 128 | 31.1 | 0.065 |
| | ≥150ml | 50 | 234 | 284 | 68.9 | |
| Time to reach full feeding | < = 7 day | 8 | 26 | 34 | 8.3 | 0.001 |
| | 8–14 day | 32 | 190 | 222 | 53.9 | |
| | 15-< = 21 day | 16 | 116 | 132 | 32 | |
| | >22 day | 5 | 19 | 24 | 5.8 | |

Neonates who received their first feed on the first day had 3.36 times (AOR = 3.36, 95% CI: (1.63–6.92)) more likely to have adequate postnatal weight gain compared to those whose first feed was on the second day or later. Preterm VLBW neonates with respiratory distress syndrome (RDS) were 69% (AOR = 0.31, 95% CI: (0.12–0.78)) less likely to have adequate post natal weight gain compared to those without RDS. See (Table 5) for details.

## Discussion

In this study, the overall proportion of preterm VLBW neonates with adequate postnatal weight gain was found to be low, with rates falling below the expected (15 g/kg/day). The independent predictors of adequate weight gain were delivery with spontaneous vaginal delivery, early initiation of feeding on the first day, neonates who had respiratory distress syndrome, and a birth z-score less than 1.29.

In this study, the overall proportion of preterm VLBW neonates with adequate postnatal weight gain was (14.6%) CI: (10.4–20.1). The findings of this study were consistent with study conducted in Tanzania 13.2% [24] and lower than the study conducted in Kenya (27.4%) [25]. The disparities identified between past studies and this investigation could be explained by differences in gestational age, birth weight, and overall health status of the neonates. Additionally, Differences in neonatal care practices and feeding protocols, including the timing and type of nutrition provided. Disparities in healthcare infrastructure and access to medical resources between the study settings may impact the effectiveness of postnatal care. Furthermore, environmental factors, such as socioeconomic status and maternal health, further contribute to these variations.

In this study, the daily average postnatal weight gain was 11.1±3.8 g/kg/d, which was in line with the study done in Nigeria (8.2±3.3 g/kg/day) [26] and falls below the recommended preterm weight gain target (15 g/kg/day) [21, 27, 28]. This may be due to limitations in the availability of high-quality, fortifiable feeds, variations in the implementation of standardized

**Table 5. Bivariable and multivariable analysis for determinants of adequate postnatal weight gain of preterm VLBW neonates, Bahir Dar city public hospitals, 2022 (N = 412).**

| Variables | Category | Postnatal weight gain | | COR (95%CI) | AOR (95%CI) |
|---|---|---|---|---|---|
| | | Adequate | Poor | | |
| Mode-of delivery | Vaginal | 48 | 220 | 2.4 (1.23, 4.68) | **2.54 (1.17, 5.54)*** |
| | C-section | 12 | 132 | 1 | 1 |
| Gestational age | Late preterm | 36 | 138 | 2.32 (1.33, 4.07) | 2.26 (0.72, 7.09) |
| | Early preterm | 24 | 214 | 1 | 1 |
| At birth z-score | > -1.29 | 34 | 78 | 4.41(2.60, 8.11) | **4.51 (1.43,14.16)*** |
| | ≤ -1.29 | 26 | 274 | 1 | 1 |
| Age at regain BW | ≤ 14 days | 48 | 188 | 3.49 (1.79, 6.79) | 2.18 (0.89, 5.38) |
| | >14 days | 12 | 164 | 1 | 1 |
| Maximum volume | < 150ml | 10 | 118 | 0.40 (0.19, 0.81) | 1.49 (0.58, 3.82) |
| | ≥ 150ml | 50 | 234 | 1 | 1 |
| Frequency of feeding | ≤ every 3hr | 56 | 258 | 5.1 (1.80, 14.45) | 2.26 (0.58, 8.89) |
| | > every 3hr | 4 | 94 | 1 | 1 |
| Initial volume | > 2ml | 18 | 36 | 4.50 (1.63, 12.42) | 2.15 (0.64, 7.22) |
| | 2ml | 35 | 263 | 1.24 (0.50, 3.08) | 0.69 (0.24, 1.92) |
| | <2ml | 7 | 53 | 1 | 1 |
| Initiation time | first day | 44 | 126 | 4.93 (2.67, 9.10) | **3.36 (1.63, 6.92)*** |
| | 2nd day | 16 | 226 | 1 | 1 |
| RDS | Yes | 7 | 133 | 0.18 (0.08, 0.43) | **0.31 (0.12, 0.78)*** |
| | No | 53 | 217 | 1 | 1 |
| Length of hospital stay | < 21 days | 24 | 140 | 1.01(0.58, 1.76) | 0.53 (0.26, 1.29) |
| | > = 22 days | 36 | 212 | 1 | 1 |

Note: A p-value is considered statically significant when p <0.05. "1" represents reference group.

* = p-value less than 0.05,

** = p-value less than 0.01 and

*** = p-value less than 0.001,

CI: Confidence Interval, COR: Crude Odds Ratio, AOR: Adjusted Odds Ratio, BW: Birth Weight, RDS: Respiratory Distress Syndrome.

growth protocols, or differences in the overall care environment. Compared with other studies, the finding of this study were lower than the study conducted in India (16.24±2.37 g/kg/d) [21, 29]. The higher weight gain observed in the Indian study may be due to more aggressive feeding regimens, advanced neonatal care practices, and better resource availability, which can enhance growth rates. Additionally, regional variations in socioeconomic factors and health-care infrastructure may impact the effectiveness of growth interventions [30]. Similarly, the growth curves of preterm VLBW neonates in this study were different from growth charts based on the intergrowth 21st and Indian preterm neonate growth standard curves [21, 30]. See F1–F8 Figs in S1 File.

Neonates born vaginally gain 2.54 times more postnatal weight per day than those born via cesarean section. The findings of this study were supported by studies conducted in London and United states [31, 32]. This may be because C-section infants loss more weight after birth and take longer to regain their birth weight than those delivered vaginally because they become more hydrated due to intravenous fluids administered to the mother before surgery [33]. Another possible explanation is that neonates who were delivered by vaginal delivery had exposure to normal flora, which is important for gut development and helps train the immune system [32]. As immunity and absorption increased, weight gain also increased.

Neonates who had their first feed initiated on the first day had 3.36 times more adequate postnatal weight gain than those whose initiation was on the second day or beyond. This finding has been demonstrated in other studies conducted in Tanzania [24]. This may be due to early initiation of feeding is associated with early maturation of the intestinal villi and tolerance to enteral feeds [32]. As the feeding progresses, this allows for higher calorie intake, enabling adequate weight gain [34].

In this study, preterm VLBW neonates with a birth z-score >-1.29 had 4.51 times more adequate postnatal weight gain compared to those with a birth z-score of ≤ -1.29. This finding was also supported by a study conducted in South Africa [35]. This may be due to a higher birth z-score reflects better intrauterine growth, which leads to more substantial nutritional reserves and improved postnatal metabolic adaptation. These factors enhance physiological resilience and facilitate more effective feeding and nutrient utilization.

This study also found an association between respiratory distress syndrome and postnatal weight gain. Preterm VLBW neonates with RDS were 69% less likely to gain adequate weight than those without RDS. Several studies conducted in Uganda, Korea, Tanzania, and Nigeria have shown that comorbid conditions in preterm VLBW neonates are compelling reasons for poor weight gain [3, 16, 36, 37]. The association between respiratory distress syndrome (RDS) and inadequate weight gain in preterm VLBW neonates may be explained by increased metabolic demands and feeding challenges associated with RDS. Elevated respiratory effort and oxygen requirement often raise the infant's metabolic rate, diverting energy from growth and recovery. In addition, RDS disrupts regular feeding patterns due to poor coordination of breathing and feeding, further limiting caloric intake. Together, these physiological demands and disrupted feeding schedules reduce the energy available for growth, leading to insufficient postnatal weight gain.

## Conclusion and recommendations

This study found that postnatal weight gain among preterm VLBW neonates in Bahir Dar city hospitals was below the expected rate of 15 g/kg/day. Key factors influencing weight gain included the mode of delivery, birth z-score, initiation time of the first feed, and the presence of respiratory distress syndrome (RDS). To address these issues, it is recommended to develop population- and gestational-age-specific growth charts for Ethiopian preterm VLBW neonates, encourage spontaneous vaginal delivery by improving labor management and training for birth attendants, prioritize maternal nutrition and health, establish protocols for initiating feeding within 24 hours of birth, provide lactation support, train healthcare staff on the importance early feeding, and implement early identification and management of RDS. Future research should focus on prospective studies to further validate these findings.

## Supporting information

**S1 File.**
(DOCX)

**S1 Questionnaire. Questionnaires on patterns of postnatal weight gain in preterm very low birth weight infants born in Bahir-Dar public hospitals, 2022, cross sectional study.**
(DOCX)

## Acknowledgments

First, we would like to express our deepest gratitude to Debre Berhan University, College of Health Sciences for supporting this research project. We would also like to extend our

appreciation to the staff of Felege Hiwot, Tibebe Gion, and Addisalem hospitals as well as to the card extractors. It is also our pleasure to express our gratitude to the data collectors.

## Author Contributions

**Conceptualization:** Yihenew Ayehu Dessie, Solomon Hailemeskel.

**Data curation:** Yihenew Ayehu Dessie, Bekahegn Girma, Solomon Hailemeskel.

**Formal analysis:** Yihenew Ayehu Dessie, Elda Mekonnen Nigussie.

**Investigation:** Yihenew Ayehu Dessie.

**Methodology:** Yihenew Ayehu Dessie, Elda Mekonnen Nigussie, Bethelehem Taye Mengistu, Leweyehu Alemaw Mengstie, Bekahegn Girma, Solomon Hailemeskel.

**Project administration:** Solomon Hailemeskel.

**Resources:** Yihenew Ayehu Dessie.

**Software:** Yihenew Ayehu Dessie, Worku Abemie, Leweyehu Alemaw Mengstie.

**Supervision:** Bethelehem Taye Mengistu, Bekahegn Girma.

**Validation:** Worku Abemie, Bekahegn Girma.

**Visualization:** Bethelehem Taye Mengistu.

**Writing – original draft:** Yihenew Ayehu Dessie, Bekahegn Girma, Solomon Hailemeskel.

**Writing – review & editing:** Yihenew Ayehu Dessie, Worku Abemie, Elda Mekonnen Nigussie, Leweyehu Alemaw Mengstie, Bekahegn Girma, Solomon Hailemeskel.

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
