## [Decision Letter · Decision Letter 0]

19 Apr 2024

PONE-D-23-19083Patterns of Postnatal Weight Gain and its Predictors among Preterm Very Low Birth Weight Neonates Born in Bahir-Dar Public Hospitals Retrospective Follow-up Study, 2022PLOS ONE

Dear Dr. YDessie,

Thank you for submitting your manuscript to PLOS ONE. After careful consideration, we feel that it has merit but does not fully meet PLOS ONE’s publication criteria as it currently stands. Therefore, we invite you to submit a revised version of the manuscript that addresses the points raised during the review process.

 The following changes are required for acceptance:

1. Please consult a statistician and revise the statistical analyses accordingly. The binomial regression analysis needs to be reported appropriately, especially the explanation on which category the aRR are referring to (APWG or PPWG, Table 5).  Further, definition of “adequate and poor weight gain” must be throughly explained as all risk factor analyses and conclusions are based on this definition. Please cite appropriate references. 

2. This is not a follow-up study. The manuscript and the title of the manuscript need to be revised accordingly.

3. The recommendations to healthcare provider do not represent the study results.  “Warming” has not been included in the analyses but recommended as prevention measure for growth failures. The same applies to “Initiating trophic feeding on the first day of life”, timely treatment of comorbidities (except for RDS), hospital stays. Instead, frequency of feeding (<= every 3 hours) seems to be an effective measure for appropriate postnatal weight gain and should be recommended. This needs to be revised accordingly. 

4. The structured questionnaire used must be included in the supplementary material.

We look forward to receiving your revised manuscript.

Kind regards,

Luisa Anna Denkel

Academic Editor

PLOS ONE

2. In the online submission form, you indicated that [The datasets analyzed in this study are available from the corresponding author upon reasonable request.]. 

Additional Editor Comments:

Dear authors,

Thank you for submitting your manuscript “Patterns of Postnatal Weight Gain and its Predictors among Preterm Very Low Birth Weight Neonates Born in Bahir-Dar Public Hospitals Retrospective Follow-up Study, 2022” to Plos One.

Yihenew Ayehu did a retrospective cohort study with more than 400 preterm infants from three hospitals (Bahir Dar public hospitals) in Bahir Dar City, Amhara Region, Ethiopia. The authors aimed to analyze preterm infants in terms of postnatal weight gain and its predictors. This is an interesting work that enriches the literature available on preterm infants in low resource settings. However, this work requires major revisions before it can be published in Plos One.

General remarks:

Please use the term „infant“ instead of “baby” throughout the entire manuscript. Please use the term “VLBW infant” or “preterm infant”, not preterm VLBW infant.

Most importantly, please consult a statistician and revise the statistical analyses accordingly.

Please use the abbreviations (VLBW) instead of the written out terms once they have been introduced.

Line 111: The structured questionnaire used must be included in the supplementary material.

Figure 1: Please revise Fig 1 (for spelling mistakes and skewed lines)

Specific remarks:

Line 49ff: This paragraph is not fully correct and needs revision. For VLBW definition, please check the bibliography, did you cite the correct references? https://pubmed.ncbi.nlm.nih.gov/3376487/. Further, VLBW is not defined by gestational age, but by birth weight only. The sentence should be revised accordingly. It is correct that very low birth weight mainly occurs for premature infants before 37 weeks of gestational age and / or intrauterine growth restriction (Stoll BJ, Adams-Chapman I: The fetus and the Neonatal Infant. Nelson Textbook of Pediatrics. 2007, Saunders, 671-711. Eighteenth). Please include the correct reference for the definition of postnatal weight gain.

Abstract: The abbreviation SVD needs to be written out before its first use.

Line 68: please correct “challenging” or “is a challenge”

Line 70: Please introduce the term “NEC”.

Line 82: Please include a reference for this statement (“Although postnatal weight gain is a common problem in preterm very low birth weight infants in resource-limited countries”).

Line 85: Please remove “in Ethiopia” and add another sentence to report that this study was conducted in a low resource country, namely Ethiopia”.

Line 88: Please add “in Bahir Dar public hospitals in Ethiopia”.

Line 95: Please cite a reference for the national guidelines of Ethiopia.

Line 96: Please introduce the abbreviation NG for NG tubes.

Line 99: This is NOT a follow-up study, but a retrospective cohort study. Please also remove the term “follow-up” from the title.

Line 100: What does the study period (June 12 – July 20, 2022) refer to? Is this the time period when the retrospective study was conducted? Please report the time period when children were born a how long they have been followed up. Did you include all preterm infants that fulfilled the inclusion criteria?

Line 106: the phrase “were died or discharged” must be corrected.

Line 111: The questionnaire used must be included in the supplementary material.

Line 115: Please cite a reference for the two-point average model used.

Line 121: How many data collectors worked for the study? I understood that this retrospective analysis was based on chart review only, no measurements of weigth were conducted by the data collectors? Is this correct? What did the teaching of data collectors imply? Please specify this.

Line 166: Please add a “dot” at the end of the sentence.

Line 175: Please explain why only 206 VLBW were included in the analyses (while 412 were selected and recruited). Please include the exclusion steps /criteria to Fig 1.

Line 176: What do the authors mean by “response rate”?

Line 177: Do the authors “Caesarian section” or “Caesarian” by “Section” Please revise in the entire manuscript.

Line 178: Please introduce the abbreviation “ANC”.

Line 191: Please revise spelling of “There”.

Table 1: Please report 0 observation for LGA (size for GA).

Table 1, Table 2, Table 4: Please include p-values for comparison of adequate and poor weight gain.

Please explain and name all references you used for your definition of “adequate and poor weight gain”. This is crucial to report as your risk factor analyses is based on this definition.

Line 223: This part belongs to the method’s section. Please revise this accordingly.

Line 263: Please check the reported average postnatal weight gain of “11.43.8 g/kg/d”.

Table 5:

• This table is hard to read / interpret. It is not clear, for which category the authors report CRR / aRR (APWG or PPWG). Please report postnatal weight gain by the following categories: appropriate weight gain (= 1), while poor weight gain (= no appropriate weight gain, referenced as “0”) or mark in the table which category represents the reference.

• RDS: Using “Yes” as reference is difficult to interpret for the reader.

Line 342: “Warming” has not been included in the analyses. How can the authors recommend this as prevention measure for growth failures? The same applies for “Initiating trophic feeding on the first day of life”, timely treatment of comorbidities (except for RDS), hospital stays. Instead, frequency of feeding (<= every 3 hours) seem to be an effective measure for appropriate postnatal weight gain.

Reviewers' comments:

Reviewer's Responses to Questions

**Comments to the Author**

1. Is the manuscript technically sound, and do the data support the conclusions?

Reviewer #1: Yes

Reviewer #2: Yes

Reviewer #3: Yes

2. Has the statistical analysis been performed appropriately and rigorously? 

Reviewer #1: No

Reviewer #2: Yes

Reviewer #3: Yes

3. Have the authors made all data underlying the findings in their manuscript fully available?

Reviewer #1: No

Reviewer #2: Yes

Reviewer #3: Yes

4. Is the manuscript presented in an intelligible fashion and written in standard English?

Reviewer #1: No

Reviewer #2: Yes

Reviewer #3: Yes

5. Review Comments to the Author

Reviewer #1: Background:

Line 49: Paraphrase it….. In such way…..”Very low birth weight (VLBW) preterm infant is defined by the World Health Organization 50 (WHO) as those infants born weighing less than 1500 grams before 37 completed weeks of gestation.”

Line 63: Flow of ideas are not coherent, such as similarly….. What does it links from the previous paragraph

Line 70: Some previously defined abbreviations/ Acronyms such as NEC

Generally, the background section is well written on the view of known scientific guidelines and books. However, the disparities and previous findings of literatures was not explored. Due to this, why the research was done is not clear. It is my suggestion to revise the background section by incorporating strong statement of the problem and justifications with its implication on further to the scientific community.

Methods:

Design: why it is follow up study? Why not Cohort? I am happy to know their difference.

The time period for the follow up study was nearly month (Even though the study incorporates 2 years charts). Could be a follow up study with this period? It seems cross-sectional chart review than follow up study? Nothing was followed by this study. How the study assured that the data were collected before the outcome occurred? For example, comorbid condition may occurred after the exacerbation of the weight loss.

Sample size calculation?????? And Sampling techniques????

Line 113: We are not sure whether daily weight was recorded.

Line 117/124: Paraphrase and punctuate appropriately.

Line 129: Why you operationalized some terms such as Growth velocity, Growth failure; non used words in the study results.

Line 144/147: It is not clear why such percentages were calculated, better to explain its advantages for the study findings.

Line 154: Why VIF used? To check confounding????

Result and discussion:

Line 171: The data reliability should be checked. I don’t think such number of twin/triple, death or discharged before 7 days among VLBW neonates in your study set up.

Line 175: What are the actual respondents, make clear. Is 199 or 206 or 412 or Etc.

Line 177/178: Sections??? Most neonates of neonates????

Line 187: Initially your study excluded 206 participants, but all your write up included all 412 charts. Please write again and make consistent writing throughout the documents.

Table 2: Is determining the time to reach full volume feeds the study’s objective? This may need another type of modeling and analysis.

Line 213: Did you think the comorbidity precedes the weight loss or the reverse?

Table 4: Is the objective of your study?

Line 223-234: Did the variables full filled the model assumption? Log-binomial model assumptions???? And model adequacy???

Line 235 to Table 5: The study come with the scientifically accepted predictors, which were already listed in guidelines and books. So, could the study have significant for clinicians and the literature communities?

Line 259: Still the study discussed opposite to the initial objectives of the study. And please calculate the confidence interval for poor and adequate proportion of weight gain Preterm VLBW neonates.

Generally, the discussion section was shallow and use inappropriate discussion techniques, for example, if the population was different, discussing a different population is inappropriate.

References: Most references used were outdated, please updated with the past five years.

Reviewer #2: While none of the results presented are surprising, it is important for country specific data regarding this important issue be available. My only concern is lumping together late and early preterm infants given that these are very different in their comorbidities and ability to tolerate enteral feeding.

Reviewer #3: Study aims are practical and useful for the population subset. Methodology and results meet the aims. The writing and presentation of data needs to be improved. References need to be re-written. Please refer to the attachment for more details.

6. PLOS authors have the option to publish the peer review history of their article (what does this mean?). If published, this will include your full peer review and any attached files.

Reviewer #1: No

Reviewer #2: No

Reviewer #3: No

---

## [Decision Letter · Decision Letter 1]

18 Sep 2024

PONE-D-23-19083R1Patterns of Postnatal Weight Gain and its Predictors among Preterm Very Low Birth Weight Neonates Born in Bahir-Dar City Public Hospitals, 2022: A Cross Sectional StudyPLOS ONE

Dear Dr. Dessie,

Thank you for submitting your revised manuscript to PLOS ONE. After careful consideration, we feel that it has merit but does not fully meet PLOS ONE’s publication criteria as it currently stands. Therefore, we invite you to submit a revised version of the manuscript that addresses the points raised during the review process.

Please revise the manuscript again according to the suggestions mady by the reviewers. Please submit the original data set to meet PLOS ONE’s publication criteria and to ensure transparancy.  

We look forward to receiving your revised manuscript.

Kind regards,

Luisa Anna Denkel

Academic Editor

PLOS ONE

Journal Requirements:

Reviewers' comments:

Reviewer's Responses to Questions

**Comments to the Author**

1. If the authors have adequately addressed your comments raised in a previous round of review and you feel that this manuscript is now acceptable for publication, you may indicate that here to bypass the “Comments to the Author” section, enter your conflict of interest statement in the “Confidential to Editor” section, and submit your "Accept" recommendation.

Reviewer #1: All comments have been addressed

Reviewer #3: (No Response)

2. Is the manuscript technically sound, and do the data support the conclusions?

Reviewer #1: Yes

Reviewer #3: Yes

3. Has the statistical analysis been performed appropriately and rigorously? 

Reviewer #1: Yes

Reviewer #3: Yes

4. Have the authors made all data underlying the findings in their manuscript fully available?

Reviewer #1: Yes

Reviewer #3: Yes

5. Is the manuscript presented in an intelligible fashion and written in standard English?

Reviewer #1: No

Reviewer #3: Yes

6. Review Comments to the Author

Reviewer #1: Still, I have concerns about the general idea flow, and the reason for the study being done was not well elaborated. Methodologically, the reliability and accuracy of the study's data are questionable.

Reviewer #3: 1. Please define “early” vs “late” preterm deliveries.

2. The “number of children that the mother has”, although relevant, does not add up to n= 412. The information does not appear to contribute to final data or discussion and may be dropped, if incomplete.

3. Although it appears to be weight loss within the first week of life, please define “average weight loss" in table 3.

4. Data for Z score at birth is flipped in table 3.

5. Please recheck p=value calculation for discharge weight category in table 3.

6. Please follow APA format for table titles.

7. Spelling and grammar re-checks needed. The letter “c” in word “city” should not be capitalized, for example. “Loss” instead of “lose” in table 3. Some other lines like 164-165 (birth weights don’t change; only weights do). Line 45, 46, 47; 134, etc.

7. PLOS authors have the option to publish the peer review history of their article (what does this mean?). If published, this will include your full peer review and any attached files.

Reviewer #1: No

Reviewer #3: No

---

## [Author Response · Author response to Decision Letter 1]

4 Nov 2024

we attached as a separate file. thank you

---

## [Editor Report · Decision Letter 2]

8 Nov 2024

PONE-D-23-19083R2Patterns of Postnatal Weight Gain and its Predictors among Preterm Very Low Birth Weight Neonates Born in Bahir-Dar City Public Hospitals, 2022: A Cross Sectional StudyPLOS ONE

Dear Dr. Dessie,

Thank you for re-submitting your manuscript to PLOS ONE. After careful consideration, we feel that it has merit but does not fully meet PLOS ONE’s publication criteria as it currently stands. Therefore, we invite you to submit another revised version of the manuscript that addresses the points raised during the review process. Thank you for your careful revision of the manuscript. Please re-revise the manuscript considering the following issues:

- Line 29: Please specify which mode of delivery (compared to what) and which z-scores were identified as “risk factors” (predictors) for postnatal weight gain. Please avoid colon (:) in the sentence.

- Please do not use the term “babies” but “infants” throughout the entire manuscript

- Line 72: VLBW has already been introduced in line 41, please use the abbreviation after the first introduction.

- Table 1: for gram, please use the abbreviation “g” instead of “gm”; Please re-check the spelling of the legend (small / large characters)

- Please re-check the manuscript for spelling mistakes, such as line 270 “Furthermore, Environmental factors […]”.

We look forward to receiving your revised manuscript.

Kind regards,

Luisa Anna Denkel

Academic Editor

PLOS ONE
---

## [Author Response · Author response to Decision Letter 2]

25 Nov 2024

we submit our response to the reviewer and editor's comments as a separate file

---

## [Editor Report · Decision Letter 3]

28 Nov 2024

Patterns of Postnatal Weight Gain and its Predictors among Preterm Very Low Birth Weight Neonates Born in Bahir-Dar City Public Hospitals, 2022: A Cross Sectional Study

PONE-D-23-19083R3

Dear Dr. Yihenew Ayehu Dessie,

We’re pleased to inform you that your manuscript has been judged scientifically suitable for publication and will be formally accepted for publication once it meets all outstanding technical requirements.

Kind regards,

Luisa Anna Denkel

Academic Editor

PLOS ONE

---

## [Editor Report · Acceptance letter]

29 Jan 2025

PONE-D-23-19083R3 

PLOS ONE

Dear Dr. Dessie, 

I'm pleased to inform you that your manuscript has been deemed suitable for publication in PLOS ONE. Congratulations! Your manuscript is now being handed over to our production team.

Kind regards, 

on behalf of

Dr. Luisa Anna Denkel 

Academic Editor

PLOS ONE